# The Combined Effects of an External Field and Novel Functional Groups on the Structural and Electronic Properties of TMDs/Ti_3_C_2_ Heterostructures: A First-Principles Study

**DOI:** 10.3390/nano13071218

**Published:** 2023-03-29

**Authors:** Siyu Zheng, Chenliang Li, Chaoying Wang, Decai Ma, Baolai Wang

**Affiliations:** 1College of Aerospace and Civil Engineering, Harbin Engineering University, Harbin 150001, China; 2School of Physics and Engineering, Sun Yat-sen University, Guangzhou 510275, China

**Keywords:** TMDs/MXenes, electronic properties, density functional theory (DFT), functional groups, biaxial strain, electric field

## Abstract

The stacking of Ti_3_C_2_ with transition metal dihalide (TMDs) materials is an effective strategy to improve the physical properties of a single material, and the tuning of the related properties of these TMDs/Ti_3_C_2_ heterostructures is also an important scientific problem. In this work, we systematically investigated the effects of an external field and novel functional groups (S, Se, Cl, Br) on the structural and electronic properties of TMDs/Ti_3_C_2_X_2_ heterostructures. The results revealed that the lattice parameters and interlayer distance of TMDs/Ti_3_C_2_ increased with the addition of functional groups. Both tensile and compressive strain obviously increased the interlayer distance of MoS_2_/Ti_3_C_2_X_2_ (X = S, Se, Cl, Br) and MoSe_2_/Ti_3_C_2_X_2_ (X = Se, Br). In contrast, the interlayer distance of MoSe_2_/Ti_3_C_2_X_2_ (X = S, Cl) decreased with increasing compressive strain. Furthermore, the conductivity of TMDs/Ti_3_C_2_ increased due to the addition of functional groups (Cl, Br). Strain caused the bandgap of TMDs to narrow, and effectively adjusted the electronic properties of TMDs/Ti_3_C_2_X_2_. At 9% compressive strain, the conductivity of MoSe_2_/Ti_3_C_2_Cl_2_ increased significantly. Meanwhile, for TMDs/Ti_3_C_2_X_2_, the conduction band edge (CBE) and valence band edge (VBE) at the M and K points changed linearly under an electric field. This study provides valuable insight into the combined effects of an external field and novel functional groups on the related properties of TMDs/Ti_3_C_2_X_2_.

## 1. Introduction

Due to the inter-layer coupling effect, heterostructures formed by vertically stacking different two-dimensional (2D) materials can achieve ultra-high-performance improvements and possess unprecedentedly excellent physical properties [1,2,3]. The transition metal dihalides (TMDs) have excellent band gap widths in the range of 1.0 eV to 2.0 eV [4]. The monolayer MoS_2_, as a typical representative member of the most studied TMD family, is a semiconductor material with a direct band gap of 1.8 eV, and it is widely used in logic transistors and photodetector devices [5,6]. Similarly, the monolayer MoSe_2_, as another important member of TMDs, has many promising applications in electronics and optoelectronics due to its unique electronic, optical, mechanical, chemical, and thermal properties [7,8]. However, TMDs also have some negative properties that affect their application. For example, their carrier effective mass is relatively high, while the carrier mobility is very low [9,10], which hinders their application in high-performance nanodevices. Several studies have shown that the heterogeneous structures formed by stacking TMDs with other 2D materials can significantly modulate the structural, electronic, and mechanical properties of TMDs. For example, the studies of Biroju et al. [11] showed that MoS_2_ stacked with a bilayer graphene heterostructure can improve electronic conductivity, electrochemical properties, and photochemical properties. Li et al. [12] systematically investigated electron density differences and band gaps of Cu_3_N/MoS_2_ heterostructures, and the results showed that the charge was mainly accumulated and consumed near the atoms, with only a small amount of charge accumulating between the layers, and the bandgap of Cu_3_N/MoS_2_ heterostructures can be efficiently tuned with the variation of the interlayer distance. Moreover, Cu_3_N/MoS_2_ heterostructures have a stable structure and excellent photoelectric properties. Celal et al. [13] investigated the structural and electronic properties of GaN/MoSe_2_ heterostructures with van der Waals (vdW) correction. An indirect band gap of 1.371 eV was obtained when the GaN monolayers were adsorbed on MoSe_2_ monolayers, while when GaN was stacked on MoSe_2_ monolayers, the indirect band gap in GaN monolayers was maintained at 0.341 eV. Based on these studies, we can conclude that combining TMDs with other two-dimensional layered materials to form heterogeneous structures may be an effective way to tune and improve the relevant properties of TMDs.

MXenes are a new and important member of the family of 2D materials discovered in recent years and are complex layered 2D material systems representing a large class of transition metal nitriles, carbides, and carbonitrides [14,15,16]. Ti_3_C_2_ is a typical transition metal carbide with multilayer metal ion adsorption behavior, with ultra-high electrical conductivity and extraordinary mechanical and electronic properties [17,18]. Ti_3_C_2_ is expected to be the most competitive material candidate in some fields, such as high-performance ultra-thin electronics and storage [19,20]. Mathis et al. [21] investigated the MXene nanosheets (Al-Ti_3_C_2_), and found they have higher quality, increased oxidation resistance, and electronic conductivity increased to 20,000 S/cm. Al-Ti_3_C_2_ is a promising electric nanodevice. Li et al. [22] systematically investigated the interfacial properties of monolayer WS_2_ in contact with a series of MXenes using first-principles calculations. The results showed that Ti_3_C_2_ couples strongly with WS_2_, leading to the metallization of monolayer WS_2_ and the formation of ideal Ohmic contacts in the vertical direction. Moreover, during the electrode fabrication process, the face-to-face stacking of 1 nm thickness MXene limits the accessibility to electrolyte ions [23,24], which hinders the electronic properties utilization of its surface. For Ti_3_C_2_, heterostructure stacking is also a valuable tool for improving the electronic properties since it can not only add the excellent properties of a single 2D material but also provide a stable gallery space [25,26,27], which could prevent face-to-face stacking of MXene. For example, Wu et al. [28] systematically investigated the energy storage and electronic properties of N-Ti_3_C_2_/NiCo_2_S_4_ heterostructures. Owing to the unique heterostructure and friendly interfacial interaction, the N-Ti_3_C_2_/NiCo_2_S_4_ heterostructure had a stable structure, low internal resistance, and excellent rate performance. Debow et al. [29] found that the strong electronic coupling between Ti_3_C_2_O*_x_* and TiO_2_ is due to their proximity; the Ti_3_C_2_O*_x_*-generated electrons are transferred into the conduction band of the TiO_2_ semiconductor over the Schottky barrier with a fast time constant of 180 fs, leading to an increase in conductivity.

Currently, there are a large number of studies on TMD and MXene heterostructures. Jing et al. [30] systematically investigated the structural and electronic properties of MoS_2_/Ti_3_C_2_T*_x_* (T = OH, F and O) heterostructures, and the results showed that the MoS_2_/Ti_3_C_2_F_2_ heterostructure is an *n*-type Schottky contact and the Schottky barrier height (SBH) is 0.73 eV, while the MoS_2_/Ti_3_C_2_O_2_ heterostructure is a *p*-type Schottky contact with an SBH of 0.33 eV. Moreover, the tensile strain can effectively adjust the position of the conduction band edge (CBE) of MoS_2_, which leads to an effective reduction of the Fermi energy level pinning and SBH, thus allowing for Ohmic contact. Guan et al. [31] studied heterostructures composed of Ti_3_C_2_T_2_ (T = O and F atoms) and metallic MoS_2_ (1T phase) for lithium-ion battery (LIB) applications. The different surface functional groups in MXenes were found to significantly alter the redox reaction of Li atoms in the Ti_3_C_2_T_2_ and 1T-MoS_2_ interfaces. The diffusion curve became significantly flattened from the bared to O and F terminated Ti_3_C_2_, with the Schottky barrier height reducing dramatically from 0.80 eV to 0.22 eV and 0.29 eV, respectively. The surface functional group O or F can remove the spatial site resistance of Li embedding by disrupting the strong interaction between the two layers while providing additional adsorption sites for Li diffusion. Li et al. [32] found that MoS_2_/Ti_2_C heterostructures have good thermoelectric and transport properties, and the applied electric field or strain can significantly improve their thermoelectric and transport properties. Xu et al. [33] found that MoSe_2_/Ti_3_C_2_T*_x_* (T = OH, O, and F) heterostructures exhibit excellent electrochemical properties at very high currents and have a large potential for sodium ion storage, which can be applied to high-performance sodium–ion batteries. Ling et al. [34] demonstrated by a first-principles method that MoS_2_/Ti_3_C_2_–OH heterostructures can enhance the catalytic activity of MoS_2_ at low sulfur vacancy concentrations. Combining the MoS_2_/Ti_3_C_2_-OH heterostructure with strain engineering can realize the potential of efficient hydrogen production. Based on these previous studies, TMDs/MXenes heterostructures can improve the related properties of single TMDs or MXenes, and the surface functional groups of the monolayer MXenes are an effective way to tune the properties of the TMDs/MXenes heterostructures. However, these studies have mainly focused on the surface functional groups O, F, and OH.

Recently, Kamysbayev et al. [35] successfully synthesized MXenes with novel functional groups (S, Se, Cl, Br, Te) capped with Ti_3_C_2_, all of which showed in-plane tensile strain, especially Ti_3_C_2_Te_2_, which had the maximum in-plane tension, showing in-plane lattice expansion of more than 18%. Lattice expansion promoted the appearance of Ti_3_C_2_ electron mobility over 10^4^ cm^2^/V·s at room temperature, as well as superconductivity. Based on the studies of Kamysbayev et al. [35], it can be concluded that these novel functional groups (S, Se, Cl, Br) obviously induce a change in the related properties of Ti_3_C_2_. Moreover, as we know, the external field can induce fancy changes in the properties of 2D materials. However, it is still unclear about the effects of these novel functional groups (S, Se, Cl, Br) and external fields on the structural and electronic properties of the TMDs/Ti_3_C_2_ heterostructures. In this paper, we first systematically investigated the structural and electronic properties of these TMDs/Ti_3_C_2_X_2_ (X = S, Se, Cl, Br) heterostructures using the density functional theory (DFT), and discussed in detail the effects of these novel surface functional groups on the related properties of these TMDs/Ti_3_C_2_ heterostructures. In addition, we then further explored the effect of external biaxial strain and electric field on the structural and electronic properties of these TMDs/Ti_3_C_2_X_2_ heterostructures.

## 2. Calculation Method

All calculations were performed within the framework of density functional theory using the CASTEP code [36]. The Perdew–Burke–Ernzerhof (PBE), based on the generalized gradient approximation (GGA), was used as the exchange–correlation function [37]. In order to accurately represent the van der Waals interactions between monolayer TMDs and 2D Ti_3_C_2_, a semi-empirical dispersion correction in Grimme format (DFT–D) was used [38]. The Ultrasoft pseudopotential [39] was used to describe the ion–electron interaction. The valence electrons performed as the [Ar]3p3d4s configuration for the Ti atom, the [Ne]3s3p configuration for the S and Cl atoms, the [Kr]4d5s configuration for the Mo atom, the [He]2s2p configuration for the C atom, and the [Ar]4p4s configuration for Br and Se atoms. The energy cutoff was set to 450 eV [40]. Structural optimization was performed using a 9 × 9 × 1 Monkhorst–Pack grid *K*-point sampling in the Brillouin zone in the unit cell, with the optimization energy convergence parameter set to 10^−5^ eV/atom and the force convergence parameter on the atoms set to 0.03 eV/Å [41]. The *K*-point grid was increased to 11 × 11 × 1 for the calculation of energy bands and density of states [42]. A vertical vacuum layer thickness of more than 15 Å was set to prevent periodic boundary interactions between adjacent layers [43]. The binding energy of the TMDs/Ti_3_C_2_ heterostructure was defined as:$$E_{b}=(E_{TMDs/Ti_{3}C_{2}}-E_{Ti_{3}C_{2}}-E_{TMDs})/A$$
where $E_{TMDs/Ti_{3}C_{2}}$, $E_{Ti_{3}C_{2}}$, and $E_{TMDs}$ are the total energies of the heterostructure, bare Ti_3_C_2_, and the TMD monolayer, respectively, and *A* is the interface area [44]. We not only investigated the structural and electronic properties of pristine TMDs/Ti_3_C_2_ and TMDs/Ti_3_C_2_X_2_ (X = S, Se, Cl, Br) but also explored the effect of biaxial strain on the structural and electronic properties of TMDs/Ti_3_C_2_ and TMDs/Ti_3_C_2_X_2_ (X = S, Se, Cl, Br) heterostructures. The biaxial strain was defined as:
*ε_x_* = (*a − a*_0_)/*a*_0_ × 100%
*ε_y_* = (*b − b*_0_)/*b*_0_ × 100%
where and *a*_0_ are the *x*-axis lattice constants in the presence and absence of strain, respectively; and *b* and *b*_0_ are the *y*-axis lattice constants in the presence and absence of strain, respectively [45]. In addition, positive (negative) values indicate tensile (compressive) strain. All heterostructure structures were relaxed.

## 3. Results and Discussions

### 3.1. Structural Properties of the TMDs/Ti_3_C_2_ Heterostructures

The lattice constants of our optimized MoS_2_, MoSe_2_, and Ti_3_C_2_ monolayers were 3.15 Å, 3.24 Å and 3.12 Å, respectively, which were in good agreement with the previous studies [46,47]. They all have a hexagonal crystal structure with a space group of P63/mmc, and they possess a lattice mismatch rate within a reasonable range of less than 4%, allowing the construction of heterostructures [48]. According to the high-symmetry stacking mode, there are six possible TMDs/Ti_3_C_2_ configurations, taking MoS_2_/Ti_3_C_2_ heterostructures as an example, see Figure 1 (six high-symmetry MoSe_2_/Ti_3_C_2_ heterostructures, see Appendix A): (a) the ZM_SA Configuration: S and Mo atoms of MoS_2_ are on top of Ti and C atoms of Ti_3_C_2_, respectively; (b) the ZM_AA Configuration: S and Mo atoms of MoS_2_ are on top of C and Ti atoms of Ti_3_C_2_, respectively; (c) the ZM_AS Configuration: S and Mo atoms of MoS_2_ are on top of Ti atoms and hollow sites of Ti_3_C_2_, respectively; (d) the MZ_SA Configuration: S and Mo atoms of MoS_2_ are on top of C atoms and hollow sites of Ti_3_C_2_, respectively; (e) the MZ_AA Configuration: Mo and S atoms of MoS_2_ are on top of C atoms and hollow sites of Ti_3_C_2_, respectively; (f) the MZ_AS Configuration: Mo and S atoms of MoS_2_ are on top of Ti atoms and hollow sites of Ti_3_C_2_, respectively.

Table 1 lists our calculated binding energies, interlayer distance, and bond lengths for the six possible stacking configurations of MoS_2_/Ti_3_C_2_ heterostructures. It can be seen that the binding energies of these MoS_2_/Ti_3_C_2_ heterostructures were all negative, indicating that the formation of all the heterostructures was exothermic. A lower binding energy represents a more stable heterostructure structure. The ZM_SA configuration of MoS_2_/Ti_3_C_2_ had the lowest binding energy of −1.79 meV/Å, so it was energetically the most stable configuration among the six configurations. It is noted that the interlayer distance *d* of the MoS_2_/Ti_3_C_2_ heterostructure was very small (in the range of 1.68 Å to 2.47 Å), which indirectly indicated a strong interaction between the layers [49,50]. Table 2 presents the binding energy of −1.03 meV/Å for the most stable configuration SA_ZM of the MoSe_2_/Ti_3_C_2_ heterostructure, while the AS_ZM configuration had the maximum binding energy of −0.39 meV/Å. The binding energies of all six configurations were negative, indicating the stability of the MoSe_2_/Ti_3_C_2_ heterostructure. The interlayer distance *d* of the six configurations of MoSe_2_/Ti_3_C_2_ heterostructures ranged from 1.89 Å to 2.56 Å, which was similar to that of MoS_2_/Ti_3_C_2_ heterostructures.

A comparison of the data in Table 1 and Table 2 showed that the binding energies of all these MoS_2_/Ti_3_C_2_ heterostructures were smaller than those of the corresponding MoSe_2_/Ti_3_C_2_ heterostructures; meanwhile, MoS_2_/Ti_3_C_2_ had a smaller interlayer distance than the MoSe_2_/Ti_3_C_2_ heterostructures. These findings showed that the MoS_2_/Ti_3_C_2_ heterostructures are more stable. We calculated the Ti–C bond length in the original monolayer Ti_3_C_2_ as 2.057 Å. As shown in Table 1 and Table 2, the Ti(3)–C(2) bond length *d*_Ti(3)–C(2)_ showed almost no change, but the Ti(1)–C(1) bond lengths of the upper layer of Ti_3_C_2_ were stretched in the TMDs/Ti_3_C_2_ heterostructures. The reason for this may be that the charge transfer from Ti_3_C_2_ to TMDs leads to the stretching of the Ti(1)–C(1) bond. The Mo–S(1) minus Mo–S(2) value (*d*_12_) and Mo–Se(1) minus Mo–Se(2) value (*d*_34_) are also presented in Table 1 and Table 2. *d*_12_ ranged from 0.021 Å to 0.134 Å, and *d*_34_ ranged from 0.013 Å to 0.107 Å, indicating that the TMDs/Ti_3_C_2_ heterostructure slightly destabilizes the TMDs monolayer. Moreover, the *d*_12_ values in the MoS_2_/Ti_3_C_2_ heterostructure were all greater than the *d*_34_ in the MoSe_2_/Ti_3_C_2_ heterostructure, suggesting that the interlayer electron coupling effect of MoS_2_/Ti_3_C_2_ is greater than that of MoSe_2_/Ti_3_C_2_. This conclusion was consistent with the results of our calculated interlayer distance and binding energy. We also noted that the binding energy and interlayer distance of the ZM_SA configurations were the smallest, so the ZM_SA configuration is the most stable configuration among the 12 configurations of MoS_2_/Ti_3_C_2_ heterostructures and MoSe_2_/Ti_3_C_2_ heterostructures considered here.

We also further investigated the effects of novel functional groups (S, Se, Cl, Br) on the structural properties of TMDs/Ti_3_C_2_ heterostructures. Previous experimental studies [51] have shown that the Ti_3_C_2_X_2_ (X = S, Se, Cl, Br) are the most stable when the surface functional groups of Ti_3_C_2_ are located in the cavity centers of Ti atoms and aligned perpendicularly to the Ti atoms in the middle layer because of the site-blocked repulsion reaction between the C atoms and the surface functional groups. Therefore, in this work, when the surface of Ti_3_C_2_ was terminated by functional groups (S, Se, Cl, Br) in MoS_2_/Ti_3_C_2_X_2_ and MoSe_2_/Ti_3_C_2_X_2_ heterostructures, we focused on the case that the functional groups are located above the hole center of the Ti atom and perpendicular to the middle Ti atom, as shown in Figure 2a,b, respectively.

It can be clearly observed from Figure 2c that the lattice parameters of TMDs/Ti_3_C_2_X_2_ heterostructures changed significantly due to the addition of surface functional groups. The lattice parameter of the MoS_2_/Ti_3_C_2_ heterostructure was 3.124 Å. With the addition of each functional group, the lattice parameter obviously increased, and MoS_2_/Ti_3_C_2_Br_2_ had the maximum lattice parameter of 3.224 Å. The lattice parameter of MoS_2_/Ti_3_C_2_S_2_, MoS_2_/Ti_3_C_2_Se_2_ and MoS_2_/Ti_3_C_2_Cl_2_ increased to 3.163 Å, 3.179 Å and 3.186 Å, respectively. Similarly, the lattice parameters of the MoSe_2_/Ti_3_C_2_ heterostructure increased with the addition of functional groups. The lattice parameters of the MoSe_2_/Ti_3_C_2_ heterostructure were 3.148 Å, while MoSe_2_/Ti_3_C_2_S_2_, MoSe_2_/Ti_3_C_2_Se_2_, MoSe_2_/Ti_3_C_2_Cl_2_, and MoSe_2_/Ti_3_C_2_Br_2_ were 3.189 Å, 3.213 Å, 3.221 Å, and 3.262 Å, respectively. Therefore, the surface functional groups had a significant effect on the structural properties of MoS_2_/Ti_3_C_2_X_2_ and MoSe_2_/Ti_3_C_2_X_2_.

To further understand the stability of these TMDs/Ti_3_C_2_X_2_ heterostructures, we calculated the binding energy, interlayer distance, and structural parameters of TMDs/Ti_3_C_2_X_2_ (X = S, Se, Cl, Br) as shown in Table 3. We provide the coordinates of the TMDs/Ti_3_C_2_X_2_ (X = S, Se, Cl, Br) optimized structure in the Appendix A. The binding energies of MoS_2_/Ti_3_C_2_X_2_ and MoSe_2_/Ti_3_C_2_X_2_ were negative. The MoSe_2_/Ti_3_C_2_Cl_2_ heterostructure possessed the highest binding energy (−3.12 meV/Å), while the MoS_2_/Ti_3_C_2_S_2_ heterostructure had the lowest binding energy (−8.44 meV/Å), indicating that the MoS_2_/Ti_3_C_2_S_2_ heterostructure was more stable. The interlayer distance of these heterostructures varied from 2.78 Å to 3.14 Å, which was significantly larger than that of the TMDs/Ti_3_C_2_ heterostructure, and a larger interlayer distance indicated a weaker electronic coupling between the monolayer TMDs and T_3_C_2_X_2_. It can be concluded that MoS_2_/Ti_3_C_2_X_2_ and MoSe_2_/Ti_3_C_2_X_2_ are typical van der Waals heterostructures.

It can also be seen from Table 3 that in MoS_2_/Ti_3_C_2_X_2_ and MoSe_2_/Ti_3_C_2_X_2_, the surface functional group of monolayer Ti_3_C_2_ caused an increase in the Ti–C bond length (*d*_Ti–C_). Among them, the *d*_Ti–C_ of the MoSe_2_/Ti_3_C_2_S_2_ heterostructure reached 2.193 Å, an increase of 0.136 Å from the original 2.057 Å. The Ti–C bond length *d*_Ti–C_ of the MoS_2_/Ti_3_C_2_Cl_2_ heterostructure increased by 0.047 Å. The bond lengths between Ti(1) and X in the top layer of Ti_3_C_2_X_2_ were almost the same as those between Ti(2) and X in the bottom layer of Ti_3_C_2_X_2_, which means that the stacking of MoS_2_ and MoSe_2_ in these heterostructures hardly changes the spatial structure of Ti_3_C_2_X_2_. The strong coupling between the Ti_3_C_2_ and TMD interface was weakened by the functional group (S, Se, Cl, Br). The value of *d*_56_ was defined as Mo–Se(1) minus Mo–Se(2) or Mo–S(1) minus Mo–S(2) in TMDs/Ti_3_C_2_X_2_ (X = S, Se, Cl, Br). The MoS_2_/Ti_3_C_2_S_2_ heterostructure possessed the maximum *d*_56_ (0.009 Å), while *d*_56_ was zero in the MoSe_2_/Ti_3_C_2_Se_2_ heterostructure, indicating that the spatial structure and two-dimensional properties of the original MoSe_2_ were well preserved. The above results show that the addition of surface functional groups can seriously weaken the electronic coupling between the monolayer TMDs and Ti_3_C_2_.

### 3.2. Electronic Properties of the TMDs/Ti_3_C_2_X_2_ Heterostructures

To investigate the electronic properties of the TMDs/Ti_3_C_2_ heterostructures, we choose the most stable configurations of ZM_SA and SA_ZM to further study the electronic properties of MoS_2_/Ti_3_C_2_X_2_ and MoSe_2_/Ti_3_C_2_X_2_, respectively. The energy band structures and density of states of the MoS_2_/Ti_3_C_2_ heterostructure are presented in Figure 3. The Fermi energy level is set at zero energy. From Figure 3a, it can be seen that some energy bands cross the Fermi energy level, indicating the metallic nature of the MoS_2_/Ti_3_C_2_ heterostructure. Because of a strongly coupled interaction between the MoS_2_ and Ti_3_C_2_ monolayer, the energy bands have been hybridized severely. The total and partial density of states of the MoS_2_/Ti_3_C_2_ heterostructure is shown in Figure 3b; it can be seen that the energy band near the Fermi level is mainly dominated by the 3*d* orbit of the Ti atom and the 4*d* orbit of the Mo atom. Remarkably, the electrons of the S 3*p* orbit in the conduction band are unusually more than that of the Ti 3*p* orbit. Therefore, the conduction band is mainly contributed by the Mo 4*d*, S 3*p*, and Ti 3*d* orbitals. The energy band structures and density of states of the MoSe_2_/Ti_3_C_2_ heterostructure are presented in Figure 4. We note that the energy band of the MoSe_2_/Ti_3_C_2_ heterostructure (see Figure 4a) has no obvious hybridization around the Fermi level, which is different from the MoS_2_/Ti_3_C_2_ heterostructure. Compared with MoS_2_/Ti_3_C_2_, MoSe_2_/Ti_3_C_2_ has more concentrated energy bands, leading to a decrease in conductivity. The total and partial density of states of the MoSe_2_/Ti_3_C_2_ heterostructure is shown in Figure 4b; we know that the energy bands near the Fermi energy level are dominated by Ti 3*d* and Mo 4*d* orbitals. The Ti 3*p* orbital and C 2*p* orbital make a slight contribution near the Fermi energy level. The conduction band is mainly formed by Ti 3*d* and Mo 4*d* orbitals, while the valence band is formed by C 2*p*, Ti 3*d*, and Mo 4*d* orbitals, and the C 2*p* orbital plays a dominant role in the MoSe_2_/Ti_3_C_2_ heterostructure. Finally, by comparing the density of states of MoS_2_/Ti_3_C_2_ with that of the MoSe_2_/Ti_3_C_2_ heterostructure, we find that the peak of Ti 3*d* and Mo 4*d* in the MoS_2_/Ti_3_C_2_ heterostructure moves towards a higher energy level.

It is well known that both MoS_2_ and MoSe_2_ are semiconductors in nature and exhibit a direct band gap at the K point in the Brillouin zone [52]. However, our calculations show that the S 3*p* orbitals, the Se 4*p* orbitals, and the Mo 4*d* orbitals in the TMDs/Ti_3_C_2_ heterostructure have a large number of electrons crossing the Fermi energy level, which is sufficient to indicate that Ti_3_C_2_ induces a significant change in the electronic properties of the TMDs monolayer.

To further understand the electronic properties, the Mulliken charge and bond populations of TMDs/Ti_3_C_2_ were calculated; see Appendix A. The interlayer electron coupling of the MoS_2_/Ti_3_C_2_ heterostructure is strong due to the overlap of electron clouds between MoS_2_ and Ti_3_C_2_ (the bond population of the S–Ti bond is 0.48 Å). The Ti_3_C_2_ in the MoS_2_/Ti_3_C_2_ heterostructure obtains 0.06 *e* from the upper MoS_2_ layer, and the Ti atom in Ti_3_C_2_ loses 1.55 *e*, and 1.50 *e* of Ti is transferred to the C atom. The bonding behavior of the MoS_2_/Ti_3_C_2_ heterostructure is covalent in nature, as indicated by the bond populations. For MoSe_2_/Ti_3_C_2_, the bond population of the Se–Ti bond has a large negative value (−0.83 Å), suggesting a weak van der Waals interaction between MoSe_2_ and Ti_3_C_2_. According to the positive and negative values of the bond population, we can conclude that the bonding behavior in the MoSe_2_/Ti_3_C_2_ heterostructure is a combination of covalent and ionic bonds.

The Mulliken charge analysis shows that more electrons are transferring from Ti_3_C_2_ to MoS_2_ in the MoS_2_/Ti_3_C_2_ heterostructure. We also noted that the Mo atom in MoSe_2_/Ti_3_C_2_ and MoS_2_/Ti_3_C_2_ obtains 0.35*e* and 0.01*e*, respectively. Moreover, the partial density of states in Figure 3b and Figure 4b shows that the density of the electrons of the Mo 4*d* orbital in the MoSe_2_/Ti_3_C_2_ heterostructure at the Fermi energy level is much higher than that in the MoS_2_/Ti_3_C_2_ heterostructure. As a result, the Mo 4*d* orbital in MoSe_2_/Ti_3_C_2_ obtains more electrons from other atoms than that in MoS_2_/Ti_3_C_2_.

The electronic properties of these heterostructures were affected by the surface termination atoms of Ti_3_C_2_, as illustrated in Figure 5 and Figure 6. Compared with MoS_2_/Ti_3_C_2_ and MoSe_2_/Ti_3_C_2_, the electronic properties of MoS_2_/Ti_3_C_2_X_2_ and MoSe_2_/Ti_3_C_2_X_2_ were obviously different. Although TMDs/Ti_3_C_2_X_2_ heterostructures still exhibited metallic behavior, the peak values of density of states (DOS) at the Fermi level were lower than those of TMDs/Ti_3_C_2_ heterostructures. Based on our calculated DOS, it is worth noting that MoS_2_ and MoSe_2_ retained the semiconductor nature in the TMDs/Ti_3_C_2_X_2_ heterostructures. This means that the presence of surface functional groups (Cl, Br, S, Se) weakened the interaction strength between TMDs and Ti_3_C_2_. Compared with the DOS of the original monolayer MoS_2_ and MoSe_2_, see Appendix A, that of MoS_2_ and MoSe_2_ in MoS_2_/Ti_3_C_2_Br_2_ and MoS_2_/Ti_3_C_2_Cl_2_ showed an upward shift of the Fermi energy level. From the partial density of states (PDOS) of MoS_2_/Ti_3_C_2_Br_2_ and MoS_2_/Ti_3_C_2_Cl_2_, it can be seen that the energy band near the Fermi level was mainly contributed to by the Ti 3*d* orbit. The Mo atom 4*d* orbital in MoS_2_/Ti_3_C_2_Br_2_ made a small charge contribution, while the Mo atom 4*d* orbital in MoS_2_/Ti_3_C_2_Cl_2_ made almost no charge contribution to the Fermi level. In addition, the atoms of the functional groups Cl and Br made almost no contribution near the Fermi level, as shown in Figure 5a,b. When the Ti_3_C_2_ surface was terminated by S and Se functional groups, the DOS of MoS_2_ and MoSe_2_ showed a downward shift of the Fermi level, while the S 3*p* and Se 4*p* orbitals made an obvious charge contribution near the Fermi level. The energy band near the Fermi level became flatter, and the effective mass of the electron was larger, so the conductivity decreased. It can be seen from the density of states of Ti_3_C_2_X_2_ (X = S, Se) that the S 3*p* and Se 4*p* orbital charges and Ti 3*d* orbital generated strong hybridization, respectively, as shown in Figure 5c,d. The metal behavior of the MoS_2_/Ti_3_C_2_X_2_ (X = Br, Cl, S, Se) heterostructure was mainly dominated by the Ti 3*d* orbital charge.

Figure 6a shows the energy band and density of states of the MoSe_2_/Ti_3_C_2_Br_2_ heterostructure. The bottom of the conduction band of MoSe_2_ moved downward and coincides with the Fermi level. At the Fermi energy level, only the Ti 3*d* orbital charge contributed. For MoSe_2_/Ti_3_C_2_Cl_2_ and MoSe_2_/Ti_3_C_2_S_2_, as shown in Figure 6b,c, the Fermi energy level was located between the top of the valence band and the bottom of the conduction band. The Ti 3*d* orbital dominated the metal properties of MoSe_2_/Ti_3_C_2_Cl_2_ and MoSe_2_/Ti_3_C_2_S_2_. For MoSe_2_/Ti_3_C_2_Se_2_, as shown in Figure 6d, the energy band structure of MoSe_2_ remained unchanged, and the Fermi level was located at the top of the valence band. The Se 4*p* and Ti 3*d* orbitals in Ti_3_C_2_Se_2_ generate strong hybridization, which dominated the conductivity of MoSe_2_/Ti_3_C_2_Se_2_. Moreover, the PDOS of MoSe_2_ and MoS_2_ showed that both S 3*p* orbitals in MoS_2_ and Se 4*p* orbitals in MoSe_2_ made no charge contribution at the Fermi energy level. When S and Se were used as the functional group terminal in Ti_3_C_2_, the S 3*p* and Se 4*p* orbital charges played a leading role in the Fermi energy level.

### 3.3. Effect of Biaxial Strain on the Structural and Electronic Properties of the TMDs/Ti_3_C_2_ Heterostructures

Strain can be used to tune the electronic properties of two-dimensional materials [53,54]. Here, we systematically investigated the effect of biaxial strain on the structural and electronic properties of TMDs/Ti_3_C_2_ and TMDs/Ti_3_C_2_X_2_. Considering a series of biaxial tensile or compressive strains *ε* in 0.01 steps from −9% to +9%, *ε* > 0 and *ε* < 0 represented the tensile and compressive strains, respectively. Figure 7a shows the variation curve of the MoS_2_/Ti_3_C_2_ interlayer distance with biaxial tensile (compressive) strain. It can be seen that the interlayer distance *d* of the MoS_2_/Ti_3_C_2_ heterostructure varies linearly with an increasing biaxial tensile or compressive strain. The interlayer distance of MoS_2_/Ti_3_C_2_ decreases to 1.56 Å when the biaxial tensile strain reaches 9%. Therefore, the biaxial tensile strain can enhance the electronic coupling strength between the monolayer MoS_2_ and Ti_3_C_2_. However, its interlayer distance gradually increases when subjected to biaxial compressive strain, and the interlayer distance increased to 1.89 Å at 9% compressive strain, indicating that the electron coupling strength between the monolayer MoS_2_ and Ti_3_C_2_ is severely weakened by the biaxial compressive strain. Figure 7b presents the variation curve of the MoSe_2_/Ti_3_C_2_ interlayer distance with the biaxial tensile (compressive) strain. The result is completely different to that of the MoS_2_/Ti_3_C_2_ heterostructure. The interlayer distance of the MoSe_2_/Ti_3_C_2_ heterostructure shows a fluctuating change with an increasing biaxial tensile or compressive strain. Based on the above Mulliken charge and bond populations analysis of MoS_2_/Ti_3_C_2_ and MoSe_2_/Ti_3_C_2_, it can be seen that the bond population of S–Ti in MoS_2_/Ti_3_C_2_ is positive, while that of Se–Ti in MoSe_2_/Ti_3_C_2_ is negative. Therefore, the reason may be that the MoSe_2_/Ti_3_C_2_ heterostructure has van der Waals interactions rather than strong electron coupling interactions.

We further investigated the effect of the biaxial tensile and compressive strain on the interlayer distance *d* of TMDs/Ti_3_C_2_X_2_ heterostructures; see Figure 8. It can be seen that under the condition of the applied strain, the interlayer distance of TMDs/Ti_3_C_2_X_2_ heterostructures presents a completely different behavior to that of MoS_2_/Ti_3_C_2_. The reason is the different electronic coupling strength between the monolayer TMDs and Ti_3_C_2_X_2_ due to the surface functional groups. The interlayer distance between MoS_2_ and Ti_3_C_2_Cl_2_ remains almost unchanged under the compressive strain, while it increases slightly with the increase in the tensile strain, as shown in Figure 8a. The interlayer distance of MoS_2_/Ti_3_C_2_S_2_ heterostructures presents a steady increase under tensile strain. When the compressive strain is in the range from 0 to 5%, its interlayer distance rapidly increases; when the compressive strain is above 5%, its interlayer distance is almost unchanged. For the MoS_2_/Ti_3_C_2_Br_2_ heterostructure, under the condition of compressive or tensile strain, the coupling strength between valence electrons of Br atoms and the 4*d* orbital of Mo atoms is weakened. As a result, the interlayer distance between MoS_2_ and Ti_3_C_2_Br_2_ increases slightly with the increase in the applied strain. Compared with MoS_2_/Ti_3_C_2_X_2_ (X = S, Br, Cl), the interlayer distance of MoS_2_/Ti_3_C_2_Se_2_ presents a significantly fluctuating change in the case of compressive strain. The reason may be that the Se atom induces the aberration of a crystal lattice due to its larger atomic radii [35]. When the compressive strain is applied, the lattice distortion becomes more pronounced, resulting in the fluctuation of the interlayer distance. For MoSe_2_/Ti_3_C_2_X_2_ (X = S, Se, Cl, Br), we also observed a similar trend under the condition of compressive strain; see Figure 8b. Therefore, we concluded that the fluctuation in the interlayer distance in MoS_2_/Ti_3_C_2_X_2_ (X = Se) and MoSe_2_/Ti_3_C_2_X_2_ is mainly attributed to the existence of the Se atom, and the displacement of the Se atom affects the structural stability of these heterostructures.

Figure 8b shows that the interlayer distance of MoSe_2_/Ti_3_C_2_X_2_ dramatically increases with the increase in the tensile strain, and the interlayer distances of MoSe_2_/Ti_3_C_2_Se_2_ and MoSe_2_/Ti_3_C_2_Br_2_ at 9% tensile strain increase to 3.23 Å and 3.26 Å, respectively. Under the compressive strain, their interlayer distance increases nonlinearly. Therefore, the applied strain weakened the interlayer interaction between MoSe_2_ and Ti_3_C_2_X_2_ (X = Se, Br). For MoSe_2_/Ti_3_C_2_S_2_ and MoSe_2_/Ti_3_C_2_Cl_2_, their interlayer distance presents the same trend under the applied strain. Under the tensile strain, the interlayer distance increases, while under the compressive strain, the interlayer distance becomes increasingly smaller. Moreover, the interlayer distance of MoSe_2_/Ti_3_C_2_S_2_ is more sensitive to the applied strain than that of MoSe_2_/Ti_3_C_2_Cl_2_. It is noted that the interlayer distance of MoSe_2_/Ti_3_C_2_S_2_ increased to 3.52 Å at 9% tensile strain, in contrast; its interlayer distance decreased to 2.91 Å at 9% compressive strain. 

To explore the effect of biaxial strain on the electronic properties of TMDs/Ti_3_C_2_ heterostructures, we first investigated the Mulliken charge of TMD/Ti_3_C_2_ heterostructure under tensile and compressive strain. Table 4 shows the Mulliken charge population of MoS_2_/Ti_3_C_2_ heterostructures, at free strain, Ti_3_C_2_ loses 0.07*e*, which is transferred to the monolayer MoS_2_. At a tensile strain of 9%, Ti_3_C_2_ loses more electrons (0.11*e*). Moreover, Mo atoms also lose 0.05 electrons when subjected to tensile strain. At a compressive strain of 9%, Ti_3_C_2_ loses fewer electrons (0.03*e*). Table 5 shows the Mulliken charge population of MoSe_2_/Ti_3_C_2_ heterostructures. At a compressive strain of 9%, MoSe_2_ loses more electrons (0.08*e*), which are transferred to Ti_3_C_2_. These results showed that the interlayer electron transfer in both MoS_2_/Ti_3_C_2_ and MoSe_2_/Ti_3_C_2_ can be well regulated by biaxial strain. The difference is that in the MoS_2_/Ti_3_C_2_ heterostructure the electrons are transferred from Ti_3_C_2_ to MoS_2_, while in the MoSe_2_/Ti_3_C_2_ heterostructure, the electrons are transferred from MoSe_2_ to Ti_3_C_2_. Moreover, in the MoS_2_/Ti_3_C_2_ heterostructure the transferring electrons increase with increasing tensile strain, while in the MoSe_2_/Ti_3_C_2_ heterostructure, they increase with increasing compressive strain.

To further investigate the effect of biaxial strain on the electronic properties of TMDs/Ti_3_C_2_ heterostructures, we calculated their energy band structures and density of states under different strains. Figure 9a presents the energy band structures of MoS_2_/Ti_3_C_2_ heterostructures under compressive strain. As the compressive strain increased, the valence band I moved down and away from the Fermi energy level, and it became flatter and flatter. The energy bands II and III at the Γ point were split, and the energy band II became more dispersed. Moreover, the energy bands III and IV were combined, and then they moved away from the Fermi energy level with increasing compressive strain. Figure 9b shows the energy band structures of MoS_2_/Ti_3_C_2_ heterostructures under tensile strain. It can be seen that the increase in the tensile strain caused the valence bands I, II, and III to move away from the Fermi energy level and become flatter. Moreover, the conduction bands IV and V moved toward the Fermi energy level, and they showed more hybridization, resulting in an increase in the conductivity of the MoS_2_/Ti_3_C_2_ heterostructure [55].

Figure 10 shows the DOS of the MoS_2_/Ti_3_C_2_ heterostructure under different biaxial strains. It was obvious that the Fermi energy level was dominated by Mo 4*d* and Ti 3*d* orbitals. Moreover, the contribution of the Mo 4*d* orbital to the conduction band lessened with increasing compressive strain, while the contribution of Mo 4*d* and Ti 3*d* orbitals to the valence band increased with the increasing tensile strain. It can also be seen that the peak value of Ti atoms in the range of the conduction band increased with increasing tensile strain, while the peak value of Mo atoms remained almost unchanged with increasing tensile strain. The peak value of Mo atoms in the range of the valence band increased with the tensile strain and moved toward the Fermi energy level. Based on our calculated DOS of the MoS_2_/Ti_3_C_2_ heterostructure, the interlayer interaction between the monolayer MoS_2_ and Ti_3_C_2_ presented different behavior under tensile and compressive strain, and the interaction strength was strengthened under tensile strain, but weakened under compressive strain.

Figure 11a gives the energy band structures of the MoSe_2_/Ti_3_C_2_ heterostructure under different compressive strains. With increasing compressive strain, the conduction bands III and IV gradually merged at the Γ point and moved away from the Fermi level. The valence band I become flatter, while the valence band II crossing the Fermi level became more tortuous, indicating a decrease in conductivity. Figure 11b displays the energy band structures of the MoSe_2_/Ti_3_C_2_ heterostructure under different tensile strains. As the tensile strain increased, the energy bands II and III were split at the Γ point, and the energy bands IV and V showed more hybridization. Moreover, the energy bands crossing the Fermi energy level became more dispersed under tensile strain, inducing an increase in conductivity. We also noted that the pseudogap of the MoSe_2_/Ti_3_C_2_ heterostructure disappeared with increasing compressive or tensile strain. Therefore, the applied strain can effectively tune the electronic properties of the TMDs/Ti_3_C_2_ heterostructures.

Figure 12 illustrates the DOS of the MoSe_2_/Ti_3_C_2_ heterostructure with different biaxial strains. The peak value of Mo 4*d* orbitals at −1.8 eV gradually decreased under compressive strain, while the peak value of Ti 3*d* orbitals at 0.9 eV increased with increasing compressive strain. Under tensile strain, the peak value of both the Ti 3*d* orbital at 0.9 eV and the Mo 4*d* orbital at 1.2 eV increased with increasing tensile strain. Moreover, the peak value of Ti 3*d* orbitals moved away from the Fermi level, while in contrast, the peak value of the Mo 4*d* orbitals moved toward the Fermi energy level. It can be inferred from the variation of the DOS of MoSe_2_/Ti_3_C_2_ heterostructure with the strain that tensile strain increases the interlayer interaction, while compressive strain weakens the interlayer interaction. This result was consistent with the MoS_2_/Ti_3_C_2_ heterostructure.

We also investigated the DOS of MoS_2_/Ti_3_C_2_X_2_ under different biaxial strains, as shown in Figure 13, Figure 14, Appendix A. Compared with MoS_2_/Ti_3_C_2_Se_2_ and MoS_2_/Ti_3_C_2_Br_2_, the external strain had a slight effect on the DOS of MoS_2_/Ti_3_C_2_Cl_2_ and MoS_2_/Ti_3_C_2_S_2_, see Appendix A. The monolayer MoS_2_ remained in its band gap when a compressive strain was applied, while its band gap gradually disappeared with increasing tensile strain. This indicated that the tensile strain can improve the interaction strength between the monolayer MoS_2_ and Ti_3_C_2_Cl_2_ (or Ti_3_C_2_S_2_). For MoS_2_/Ti_3_C_2_Se_2_ and MoS_2_/Ti_3_C_2_Br_2_, the tensile strain induced the Mo 4*d* orbital to cross the Fermi level, indicating that the semiconductor nature of the monolayer MoS_2_ was completely destroyed. This means more electrons were transferred from the monolayer Ti_3_C_2_Br_2_ (or Ti_3_C_2_Se_2_) to MoS_2_. We also noted that the S 3*p* orbital in MoS_2_/Ti_3_C_2_Br_2_ obviously crossed the Fermi level when the compressive strain reached 9%. In contrast, compressive strain cannot result in the S 3*p* and Mo 4*d* orbitals crossing the Fermi level in MoS_2_/Ti_3_C_2_Se_2_, suggesting that MoS_2_ can preserve its band gap under compressive strain, while the position of its conduction band minimum (CBM) and valence band maximum (VBM) will be obviously shifted.

For comparison, the DOS of MoSe_2_/Ti_3_C_2_X_2_ under different biaxial strains was also investigated, see Figure 15, Figure 16, Appendix A. For these four kinds of MoSe_2_/Ti_3_C_2_X_2_ heterostructures, the band gap of monolayer MoSe_2_ disappeared with increasing compressive or tensile strain, and the Mo 4*d* orbitals and Se 4*p* orbitals passed through the Fermi energy levels. Therefore, the MoSe_2_/Ti_3_C_2_X_2_ heterostructure is more sensitive to external strain than the MoS_2_/Ti_3_C_2_X_2_ heterostructure. Moreover, these MoSe_2_/Ti_3_C_2_X_2_ heterostructures had a similar response to external strain. When there was no strain, the single-layer MoSe_2_ maintained its original semiconductor properties (see Figure 6), and the Fermi energy level was at the VBM of the MoSe_2_. When the tensile or compressive strain was above 6%, MoSe_2_ was transformed into a conductor. The tensile and compressive strain can obviously increase the charge contribution of Mo 4*d* orbitals and Se 4*p* orbitals at the Fermi energy level. Moreover, under compressive strain, the PDOS peak value of these MoSe_2_/Ti_3_C_2_X_2_ heterostructures becomes sharper, indicating that the electron localization is strong.

### 3.4. Effect of Electric Field on the Electronic Properties of the TMDs/Ti_3_C_2_X_2_ Heterostructures 

Some studies [56,57] have shown that the electric field is a useful way to tune the electronic properties of heterostructures. In this section, we focus on the effects of the vertical electric field on the electronic properties of TMDs/Ti_3_C_2_X_2_ heterostructures. To determine the effect of different directions of the applied electric field on the electronic properties of these heterostructures, we defined that the direction of the TMDs pointing to Ti_3_C_2_X_2_ was the positive direction of the applied vertical electric field, and the reverse direction was negative. The gradient from −0.9 V/Å to +0.9 V/Å for the applied electric field was taken in steps of 0.3 V/Å.

Figure 17 shows the DOS of MoS_2_/Ti_3_C_2_X_2_ and PDOS of MoS_2_ in MoS_2_/Ti_3_C_2_X_2_ (X = S, Se, Cl, Br) under different electric fields. The monolayer MoS_2_ maintained its band gap under the positive and negative electric fields. The position of the valence band maximum (VBM) and conduction band minimum (CBM) of MoS_2_ in MoS_2_/Ti_3_C_2_Br_2_ and MoS_2_/Ti_3_C_2_Cl_2_ remained almost unchanged under the negative electric field. In contrast, the VBM and CBM of MoS_2_ in MoS_2_/Ti_3_C_2_Br_2_ and MoS_2_/Ti_3_C_2_Cl_2_ underwent a significant shift under the positive electric field, and the peak of the Mo 4*d* orbital moved toward a higher energy level, as shown in Figure 17a,b. Compared with MoS_2_/Ti_3_C_2_Br_2_ and MoS_2_/Ti_3_C_2_Cl_2_, the effect of the electric field on the VBM and CBM of MoS_2_ in MoS_2_/Ti_3_C_2_Se_2_ and MoS_2_/Ti_3_C_2_S_2_ presented a completely reverse case. The position of VBM and CBM of MoS_2_ in MoS_2_/Ti_3_C_2_Se_2_ and MoS_2_/Ti_3_C_2_S_2_ remained almost unchanged under the positive electric field, while obviously moving under the negative electric field. Moreover, the peak position of the Mo 4*d* orbital remained almost constant, see Figure 17c,d. The DOS values of MoSe_2_ in MoSe_2_/Ti_3_C_2_X_2_ (X = S, Se, Cl, Br) under different electric fields were investigated, and the results were very similar to the MoS_2_/Ti_3_C_2_X_2_, see Appendix A. Compared with MoS_2_/Ti_3_C_2_Br_2_ and MoS_2_/Ti_3_C_2_Se_2_, the total DOS of MoS_2_/Ti_3_C_2_S_2_ and MoS_2_/Ti_3_C_2_Cl_2_ obviously moved near the Fermi Level, suggesting the shift of VBE and CBE of MoS_2_. Moreover, under the condition of a positive electric field, the total DOS of MoS_2_/Ti_3_C_2_Cl_2_ presented a sharp peak around 1 eV.

We further analyzed the energy band structure of TMDs/Ti_3_C_2_X_2_ (X = S, Se, Cl, Br) heterostructures near the Fermi energy level in the electric field range from −0.9 V/Å to 0.9 V/Å, as shown in Figure 18. Figure 18a gives the variation patterns of the energy band edges at the M and K points near the Fermi energy level for MoS_2_/Ti_3_C_2_X_2_ (X = Se, Br) heterostructures under different electric fields. For MoS_2_/Ti_3_C_2_Br_2_, the conduction band edge (CBE) and valence band edge (VBE) at both the M the K points moved towards a higher energy level with increasing electric field strength in the positive direction, while they were almost pinned under the negative electric field. In contrast, both the CBE and the VBE of MoS_2_/Ti_3_C_2_Se_2_ at the K point moved towards a higher energy level with increasing negative electric field strength, while the CBE at the K point under the positive electric field moved towards a lower energy level. Under the positive electric field, both the CBE and VBE of MoS_2_/Ti_3_C_2_Se_2_ at the M point remained unchanged. For MoS_2_/Ti_3_C_2_X_2_ (X = S, Cl), see Figure 18b, the VBE and CBE of MoS_2_/Ti_3_C_2_Cl_2_ at the M point and K point near the Fermi level were similar to those of the MoS_2_/Ti_3_C_2_Br_2_ heterostructure. The energy level of both the CBE and VBE of MoS_2_/Ti_3_C_2_S_2_ at the M point increased linearly with increasing positive electric field strength, while it decreased slightly with the negative electric field. It is worth noting that the CBE and VBE of MoS_2_/Ti_3_C_2_S_2_ at the K point were almost pinned under the positive and negative electric fields. For MoSe_2_/Ti_3_C_2_Br_2_, the energy level of the CBE and VBE at the M and K points decreased linearly with an increasing positive electric field, while it increased with an increasing negative electric field, see Figure 18c. The energy band edge of MoSe_2_/Ti_3_C_2_Se_2_ was almost independent of the positive and negative electric fields. From Figure 18d, we know that for MoSe_2_/Ti_3_C_2_S_2_ the energy level of the CBE and VBE at the K point and M point were insensitive to the external electric field. For MoSe_2_/Ti_3_C_2_Cl_2_, the energy level of the CBE at the K point and M point showed a slight change under the condition of the positive electric field, while it could be pinned when the negative electric field was applied. The energy level of the VBE at the K point and M point remained unchanged when the electric field strength was increased to 0.3 V/Å. The energy band edges of TMDs/Ti_3_C_2_X_2_ (X = S, Se, Cl, Br) heterostructures showed a significant change near the Fermi energy level under different directional electric fields, indicating that the combined functional group with the electric field can effectively tune the related properties of TMDs/Ti_3_C_2_X_2_ (X = S, Se, Cl, Br) heterostructures.

## 4. Conclusions

The effects of biaxial strain and functional groups as well as electric fields on the structural and electronic properties of TMDs/Ti_3_C_2_ heterostructures were systematically investigated based on the density functional theory method. The six possible configurations of MoS_2_/Ti_3_C_2_ and MoSe_2_/Ti_3_C_2_ heterostructure stacks were first designed, and then geometrically optimized. ZM_SA were identified as the most energetically stable structural types of MoS_2_/Ti_3_C_2_ heterostructures with a binding energy of −1.79 meV/Å^2^. The most energetically stable structure type of MoSe_2_/Ti_3_C_2_ heterostructures was SA_ZM, its binding energy of −1.03 meV/Å^2^. The surface functional groups (S, Se, Cl, Br) of the monolayer Ti_3_C_2_ resulted in the lattice expansion of TMDs/Ti_3_C_2_X_2_ heterostructures, and the MoSe_2_/Ti_3_C_2_Br_2_ heterostructure possessed the maximum lattice parameters (3.262 Å). The conductivity of MoS_2_/Ti_3_C_2_ and MoSe_2_/Ti_3_C_2_ can be enhanced by increasing the biaxial tensile strain. When the surface of the monolayer Ti_3_C_2_ was occupied by S, Se, Cl, or Br, the coupling strength between the monolayer TMDs and Ti_3_C_2_ was obviously weakened, while the biaxial strain effectively improved their interaction strength. The different surface functional groups induced a different response of the electronic properties of TMDs/Ti_3_C_2_X_2_ heterostructures to the external electric field. The energy bands around the Fermi energy level of TMDs/Ti_3_C_2_X_2_ heterostructures obviously changed under the combined effect of surface functional groups with an electric field. These results demonstrated that TMDs/Ti_3_C_2_X_2_ (X = S, Se, Cl, Br) heterostructures possess rich electronic properties. Moreover, both MoS_2_/Ti_3_C_2_X_2_ (X = Se, Br) and MoSe_2_/Ti_3_C_2_X_2_ (X = S, Se, Cl, Br) are rather sensitive to an external strain field, while only TMDs/Ti_3_C_2_X_2_ (X = Cl, Br) strongly depends on an external positive electric field. We hope that these studies can provide a theoretical foundation for the application of TMDs/MXenes heterostructures in the field of high-performance nanoelectronic devices.

## Figures and Tables

**Figure 1 nanomaterials-13-01218-f001:**
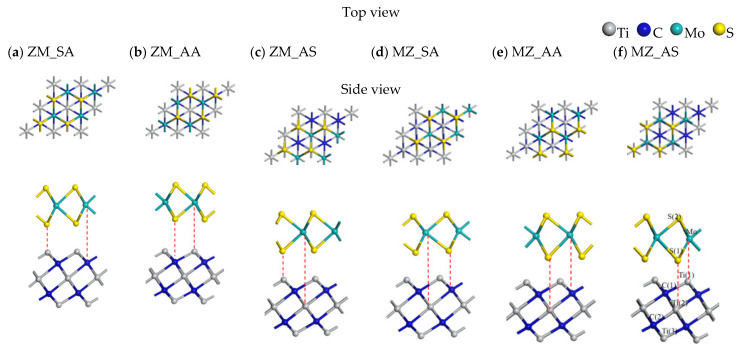
A schematic diagram of top and side views of MoS_2_/Ti_3_C_2_ heterostructures for different stackings. (**a**) the ZM_SA Configuration; (**b**) the ZM_AA Configuration; (**c**) the ZM_AS Configuration; (**d**) the MZ_SA Configuration; (**e**) the MZ_AA Configuration; (**f**) the MZ_AS Configuration.

**Figure 2 nanomaterials-13-01218-f002:**
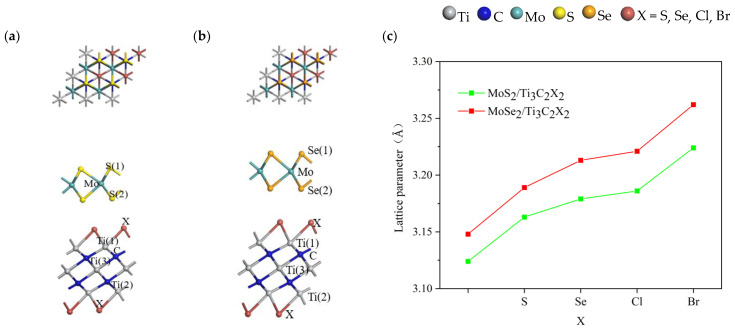
Structural properties of TMDs/Ti_3_C_2_X_2_ heterostructures. (**a**) Diagram of top and side views of MoSe_2_/Ti_3_C_2_X_2_ heterostructures; (**b**) Diagram of top and side views of MoS_2_/Ti_3_C_2_X_2_ heterostructures; (**c**) The lattice parameters of TMDs/Ti_3_C_2_X_2_ heterostructures.

**Figure 3 nanomaterials-13-01218-f003:**
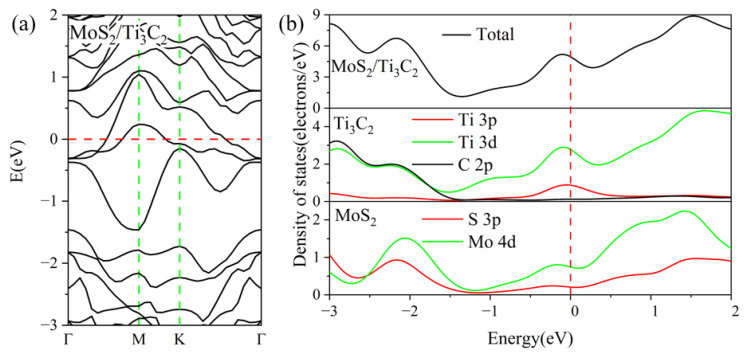
(**a**) the band structure, and (**b**) the total and partial density of states of MoS_2_/Ti_3_C_2_. The Fermi level is set to 0 eV.

**Figure 4 nanomaterials-13-01218-f004:**
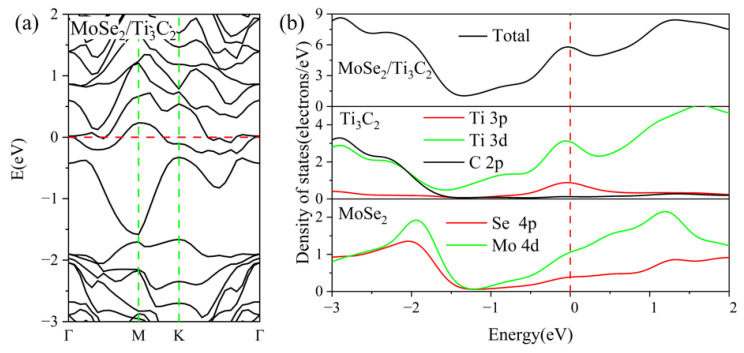
(**a**) the band structure, and (**b**) the total and partial density of states of MoSe_2_/Ti_3_C_2_. The Fermi level is set to 0 eV.

**Figure 5 nanomaterials-13-01218-f005:**
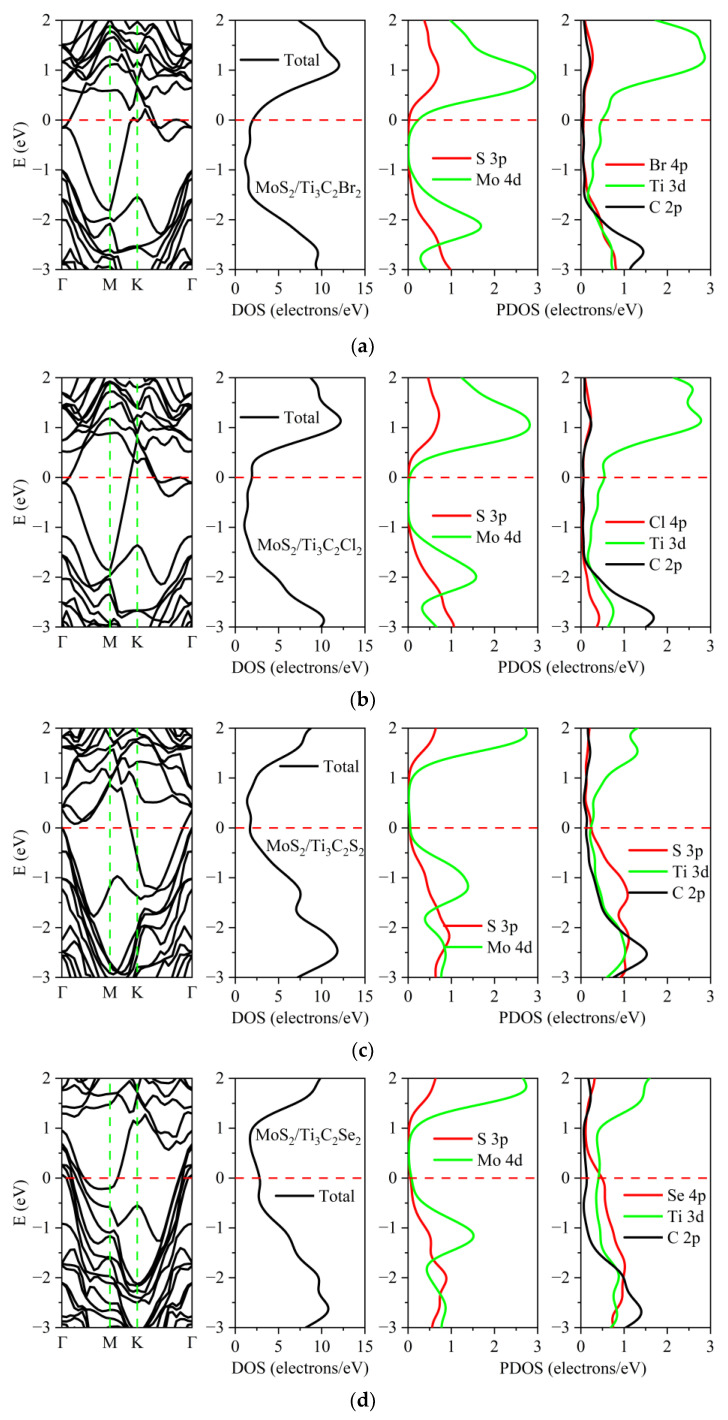
Band structures and densities of states of the MoS_2_/Ti_3_C_2_ heterostructure with different terminated groups: (**a**) -Br, (**b**) -Cl, (**c**) -S and (**d**) -Se. The Fermi level is set to 0 eV. The vertical dashed line gives the location of the Fermi level. The red line represents the Fermi level. The green line represents the high symmetry point of the Brillouin zone.

**Figure 6 nanomaterials-13-01218-f006:**
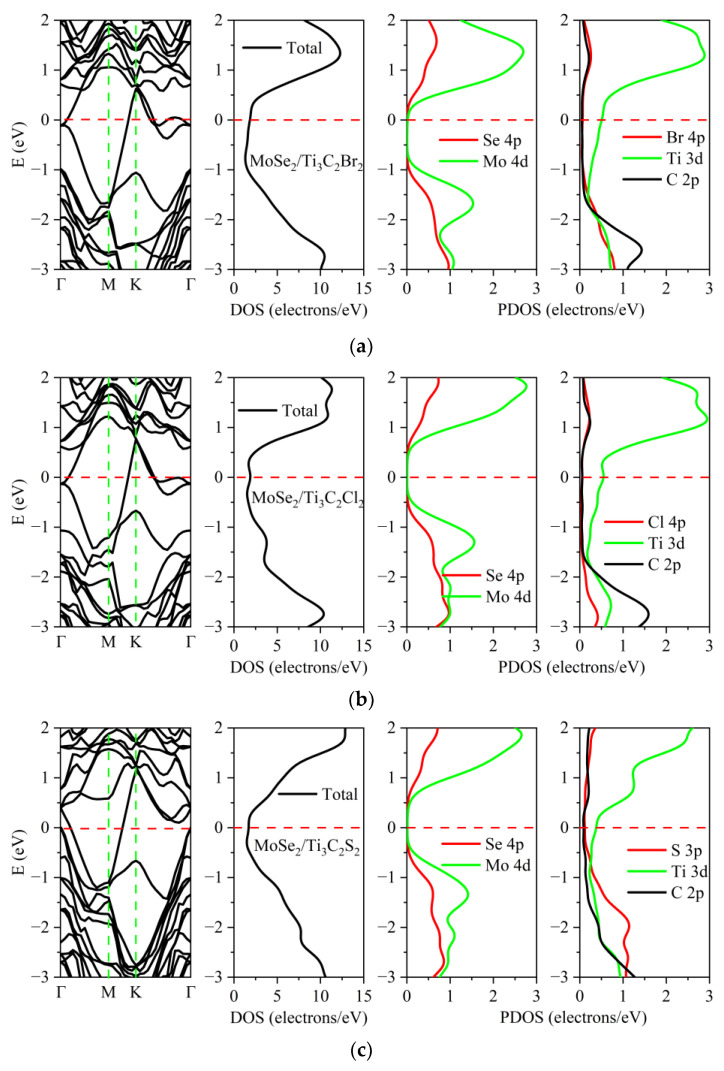
Band structures and densities of states of the MoSe_2_/Ti_3_C_2_ heterostructure with different terminated groups: (**a**) -Br, (**b**) -Cl, (**c**) -S and (**d**) -Se. The Fermi level is set to 0 eV. The vertical dashed line gives the location of the Fermi level.

**Figure 7 nanomaterials-13-01218-f007:**
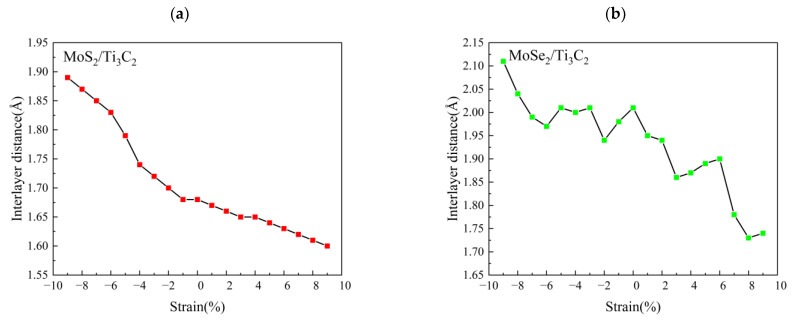
(**a**) Relationship curve between MoS_2_/Ti_3_C_2_ interlayer distance and biaxial strain, (**b**) Relationship curve between MoSe_2_/Ti_3_C_2_ interlayer distance and biaxial strain.

**Figure 8 nanomaterials-13-01218-f008:**
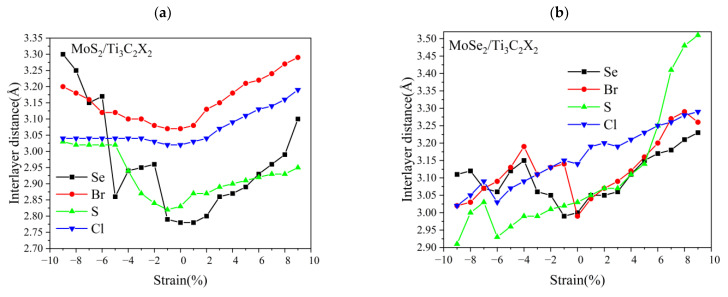
(**a**) Relationship curve between MoS_2_/Ti_3_C_2_X_2_ (X = Se, Br, S, Cl) interlayer distance and biaxial strain, (**b**) Relationship curve between MoSe_2_/Ti_3_C_2_X_2_ (X = Se, Br, S, Cl) interlayer distance and biaxial strain.

**Figure 9 nanomaterials-13-01218-f009:**
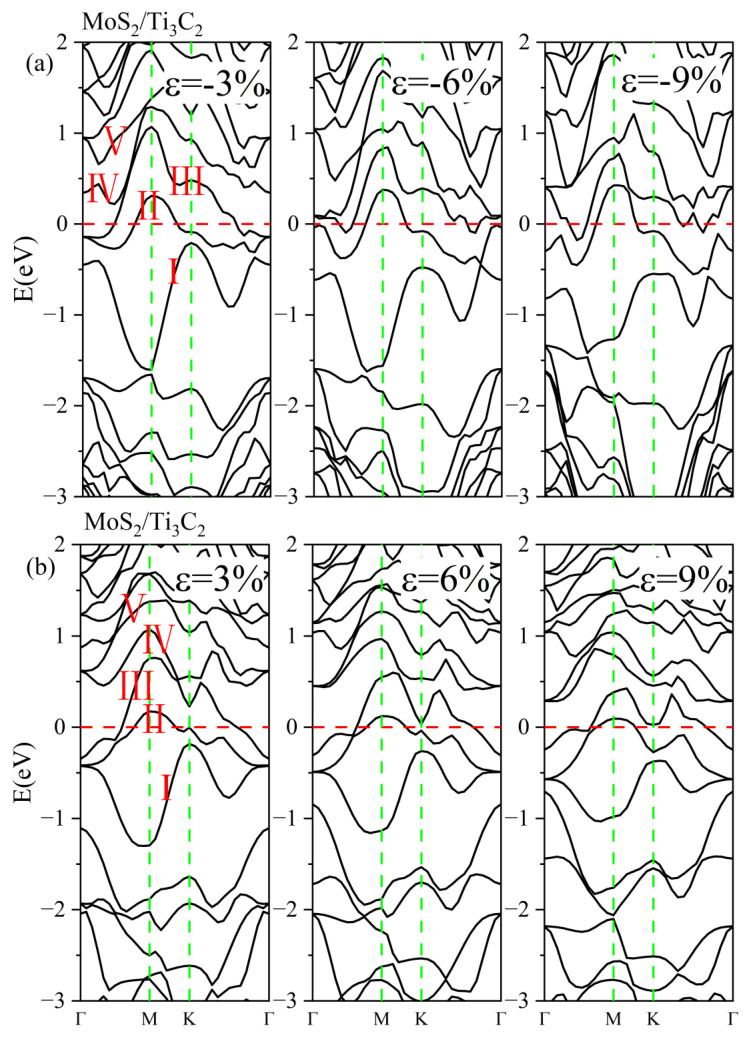
(**a**) Energy band structures of MoS_2_/Ti_3_C_2_ heterostructure under different compress strains. (**b**) Energy band structures of MoS_2_/Ti_3_C_2_ heterostructure under different tensile strains. The Fermi level is set to 0 eV. The red line represents the Fermi level. The green line represents the high symmetry point of the Brillouin zone. I/II/III/IV/V represents different energy bands.

**Figure 10 nanomaterials-13-01218-f010:**
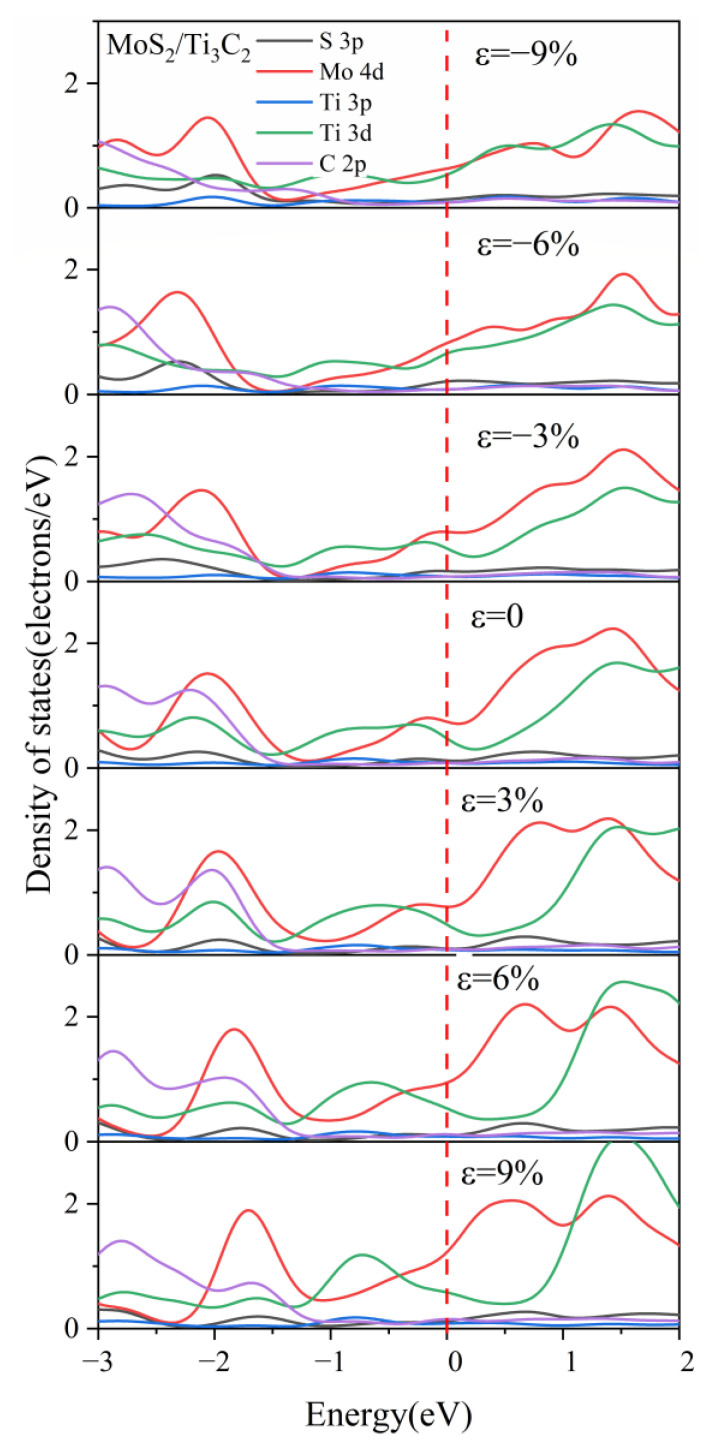
The partial density of states (PDOS) of MoS_2_ and Ti_3_C_2_ for the MoS_2_/Ti_3_C_2_ heterostructure with different biaxial strains. The Fermi level is set to 0 eV.

**Figure 11 nanomaterials-13-01218-f011:**
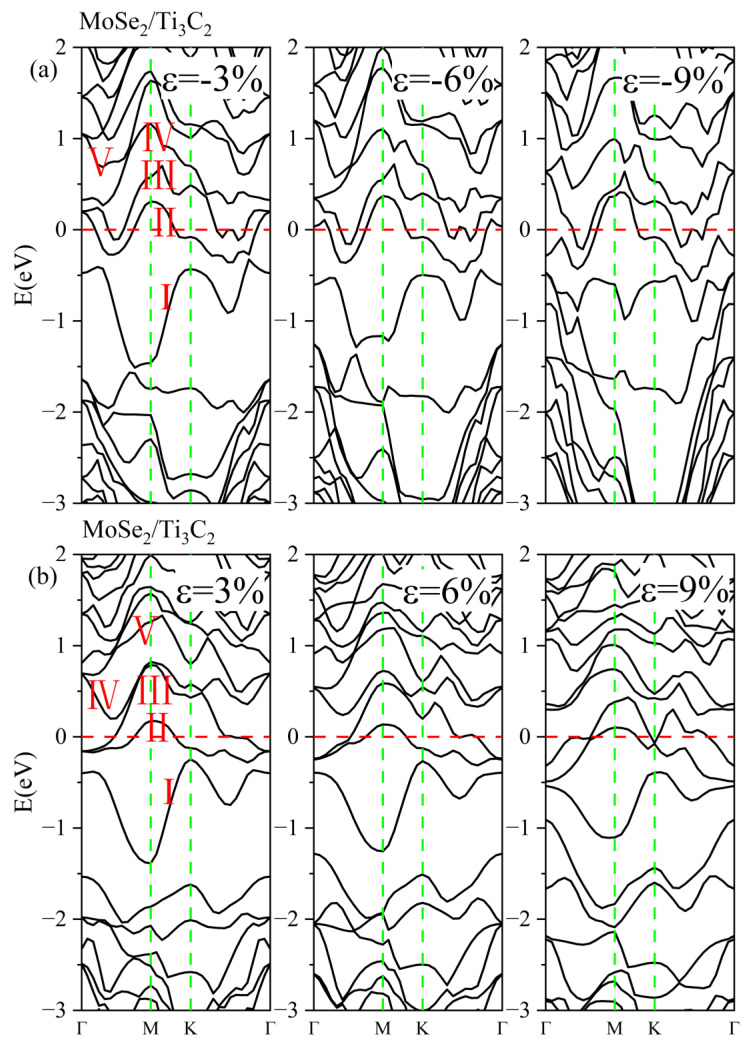
(**a**) Energy band structures of MoSe_2_/Ti_3_C_2_ heterostructure under different compress strains. (**b**) Energy band structures of MoSe_2_/Ti_3_C_2_ heterostructure under different tensile strains. The Fermi level is set to 0 eV. The red line represents the Fermi level. The green line represents the high symmetry point of the Brillouin zone. I/II/III/IV/V represents different energy bands.

**Figure 12 nanomaterials-13-01218-f012:**
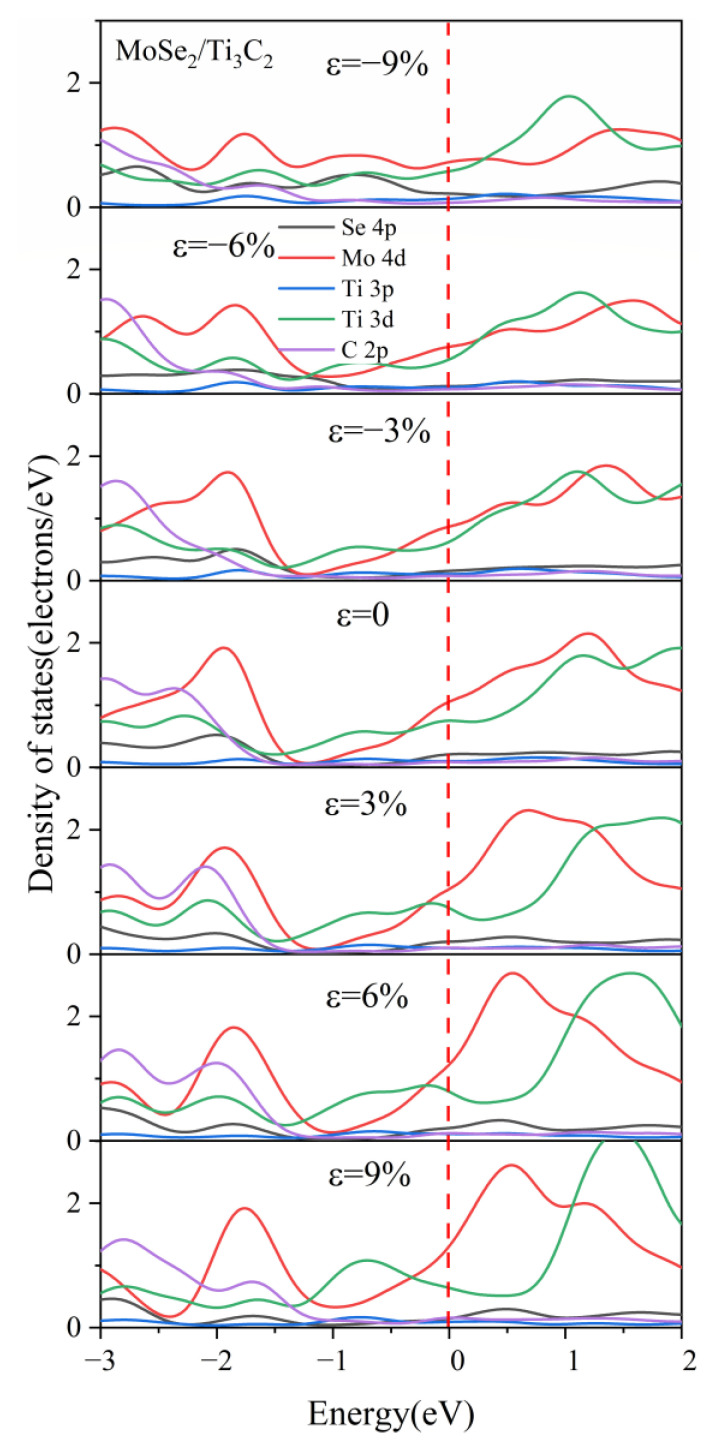
The partial density of states (PDOS) of MoSe_2_ and Ti_3_C_2_ for the MoSe_2_/Ti_3_C_2_ heterostructure with different biaxial strains. The Fermi level is set to 0 eV. The red line represents the Fermi level.

**Figure 13 nanomaterials-13-01218-f013:**
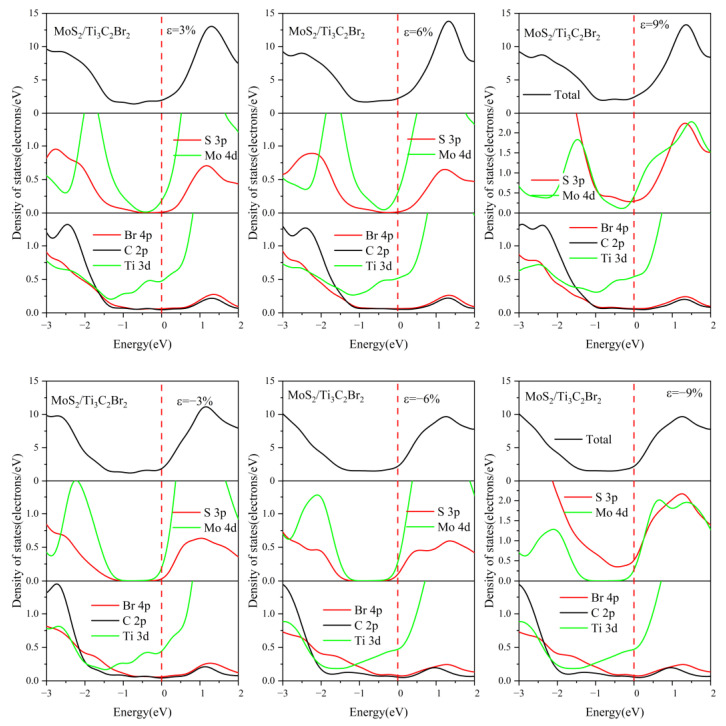
Density of states of the MoS_2_/Ti_3_C_2_Br_2_ heterostructure with different biaxial strains. The Fermi level is set to 0 eV.

**Figure 14 nanomaterials-13-01218-f014:**
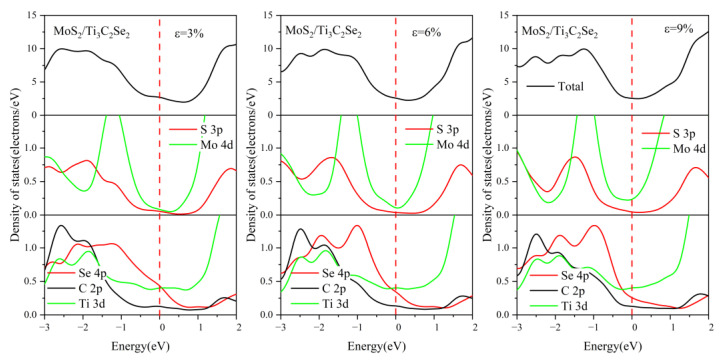
Density of states of the MoS_2_/Ti_3_C_2_Se_2_ heterostructure with different biaxial strains. The Fermi level is set to 0 eV.

**Figure 15 nanomaterials-13-01218-f015:**
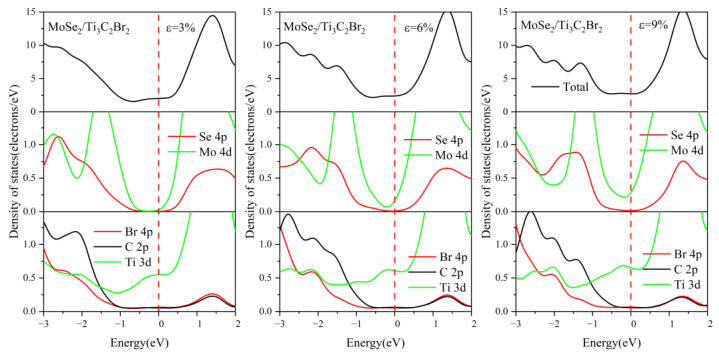
Density of states of the MoSe_2_/Ti_3_C_2_Br_2_ heterostructure with different biaxial strains. The Fermi level is set to 0 eV.

**Figure 16 nanomaterials-13-01218-f016:**
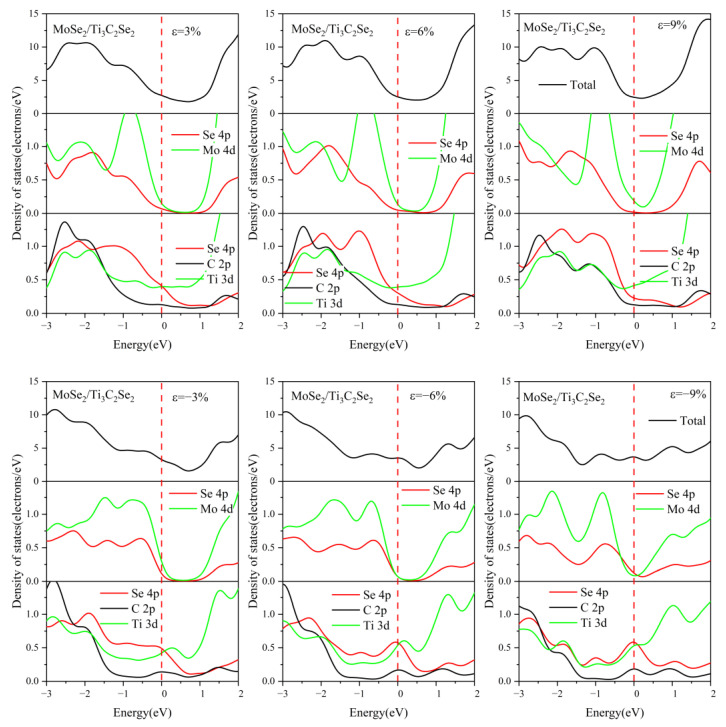
Density of states of the MoSe_2_/Ti_3_C_2_Se_2_ heterostructure with different biaxial strains. The Fermi level is set to 0 eV.

**Figure 17 nanomaterials-13-01218-f017:**
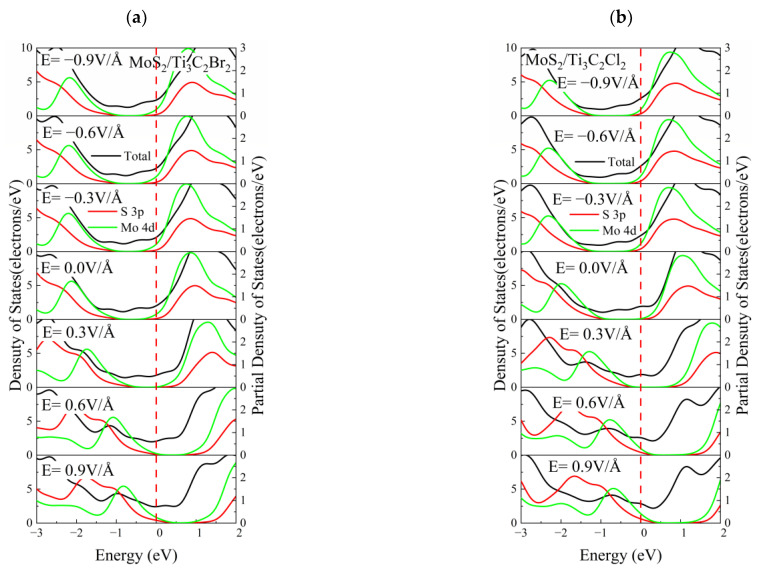
The DOS of MoS_2_/Ti_3_C_2_X_2_ and PDOS of MoS_2_ in MoS_2_/Ti_3_C_2_X_2_ heterostructures under different electric fields. (**a**) X-Br, (**b**) X-Cl, (**c**) X-Se and (**d**) X-S.

**Figure 18 nanomaterials-13-01218-f018:**
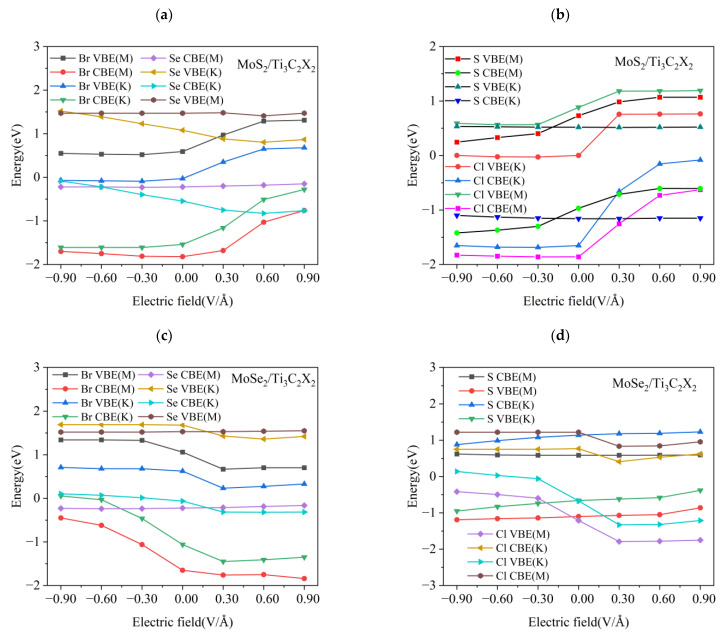
(**a**) The VBE and CBE at M and K points of MoS_2_/Ti_3_C_2_Br_2_ and MoS_2_/Ti_3_C_2_Se_2_ under different electric fields. (**b**) The VBE and CBE at M and K points of MoS_2_/Ti_3_C_2_S_2_ and MoS_2_/Ti_3_C_2_Cl_2_ under different electric fields. (**c**) The VBE and CBE at M and K points of MoSe_2_/Ti_3_C_2_Se_2_ and MoSe_2_/Ti_3_C_2_Br_2_ under different electric fields. (**d**) The VBE and CBE at M and K points of MoSe_2_/Ti_3_C_2_S_2_ and MoSe_2_/Ti_3_C_2_Cl_2_ under different electric fields.

**Table 1 nanomaterials-13-01218-t001:** Binding energy (E*_b_*), interlayer distance (*d*), Mo–S(1) minus Mo–S(2) constant (*d*_12_), and bond lengths Ti–C (*d*_Ti–C_) for MoS_2_/Ti_3_C_2_ heterostructures, respectively.

Configurations	E*_b_* (meV/Å^2^)	*d* (Å)	*d*_12_ (Å)	*d*_Ti(3)–C(2)_ (Å)	*d*_Ti(1)–C(1)_ (Å)
(a) ZM_SA	−1.79	1.68	0.069	2.059	2.125
(b) ZM_AA	−1.21	1.69	0.078	2.057	2.152
(c) ZM_AS	−1.42	2.47	0.031	2.058	2.069
(d) MZ_SA	−1.27	2.03	0.109	2.055	2.094
(e) MZ_AA	−1.48	1.92	0.134	2.057	2.081
(f) MZ_AS	−1.55	2.46	0.021	2.063	2.067

**Table 2 nanomaterials-13-01218-t002:** Binding energy (E*_b_*), interlayer distance (*d*), Mo–Se(1) minus Mo–Se(2) constant (*d*_34_), and bond lengths Ti–C (*d*_Ti–C_) for MoSe_2_/Ti_3_C_2_ heterostructures, respectively.

Configurations	E*_b_* (meVÅ^2^)	*d* (Å)	*d*_34_ (Å)	*d*_Ti(3)–C(2)_ (Å)	*d*_Ti(1)–C(1)_ (Å)
(g) SA_ZM	−1.03	1.89	0.051	2.064	2.119
(h) AA_ZM	−1.01	2.56	0.013	2.071	2.073
(i) AS_ZM	−0.39	2.51	0.021	2.067	2.074
(j) SA_MZ	−0.76	2.12	0.101	2.061	2.096
(k) AA_MZ	−0.96	2.09	0.107	2.068	2.083
(m) AS_MZ	−0.59	2.01	0.063	2.069	2.122

**Table 3 nanomaterials-13-01218-t003:** Binding energy (E*_b_*), interlayer distance (*d*), Mo–S(1) minus Mo–S(2) or Mo–Se(1) minus Mo–Se(2) constant (*d*_56_), and bond lengths Ti–X *d*_Ti–X_ and Ti–C *d*_Ti–C_ for TMDs/Ti_3_C_2_X_2_ (X = S, Se, Br, Cl) heterostructures, respectively.

Configurations	E*_b_* (meVÅ^2^)	*d* (Å)	*d*_56_ (Å)	*d*_Ti(1)–C_ (Å)	*d*_Ti(2)–X_ (Å)	*d*_Ti(1)–X_ (Å)
MoS_2_/Ti_3_C_2_S_2_	−8.44	2.83	0.009	2.182	2.397	2.389
MoS_2_/Ti_3_C_2_Se_2_	−3.95	2.78	0.007	2.146	2.520	2.496
MoS_2_/Ti_3_C_2_Cl_2_	−3.17	3.02	0.003	2.104	2.503	2.492
MoS_2_/Ti_3_C_2_Br_2_	−8.42	3.07	0.004	2.117	2.626	2.621
MoSe_2_/Ti_3_C_2_S_2_	−3.55	3.03	0.003	2.193	2.501	2.403
MoSe_2_/Ti_3_C_2_Se_2_	−4.82	3.00	0.000	2.170	2.628	2.542
MoSe_2_/Ti_3_C_2_Cl_2_	−3.12	3.14	0.003	2.120	2.509	2.506
MoSe_2_/Ti_3_C_2_Br_2_	−8.41	2.99	0.001	2.127	2.638	2.617

**Table 4 nanomaterials-13-01218-t004:** Mulliken charge (electron) of MoS_2_/Ti_3_C_2_.

Species	Ion	Total	Charge	Total	Charge	Total	Charge
		Compressive 9%		Strain Free		Tensile 9%	
C	1	4.70	−0.70	4.75	−0.75	4.78	−0.78
C	2	4.69	−0.69	4.71	−0.71	4.71	−0.71
S	1	5.94	0.06	5.98	0.02	6.05	−0.05
S	2	6.01	−0.01	6.06	−0.06	6.11	−0.11
Ti	1	11.24	0.76	11.25	0.75	11.34	0.66
Ti	2	11.62	0.38	11.59	0.41	11.57	0.43
Ti	3	11.72	0.28	11.63	0.37	11.49	0.51
Mo	1	14.08	−0.08	14.03	−0.03	13.95	0.05

**Table 5 nanomaterials-13-01218-t005:** Mulliken charge (electron) of MoSe_2_/Ti_3_C_2_.

Species	Ion	Total	Charge	Total	Charge	Total	Charge
		Compressive 9%		Strain Free		Tensile 9%	
C	1	4.69	−0.69	4.71	−0.71	4.71	−0.71
C	2	4.70	−0.70	4.75	−0.75	4.78	−0.78
Se	1	5.87	0.13	5.86	0.14	5.90	0.10
Se	2	5.68	0.32	5.77	0.23	5.74	0.26
Ti	1	11.27	0.73	11.27	0.73	11.36	0.64
Ti	2	11.71	0.29	11.62	0.38	11.49	0.51
Ti	3	11.71	0.29	11.67	0.33	11.69	0.31
Mo	1	14.37	−0.37	14.35	−0.35	14.33	−0.33

## Data Availability

Data are available on request from the corresponding author.

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
