# Peer review of "The Combined Effects of an External Field and Novel Functional Groups on the Structural and Electronic Properties of TMDs/Ti3C2 Heterostructures: A First-Principles Study"

_nanomaterials, 2023, doi:10.3390/nano13071218_

Round 1

Reviewer 1 Report

Interfaces MoSe2/Ti3C2, MoS2/Ti3C2 and TMDs/Ti3C2X2(X = S, Se, Cl, Br) for the same TMDs are modeled in this work. The DFT study of the interfaces is done. These nanomaterials are promising for batteries, catalysts, capacitors and other electrochemical applications [Adv. Sci. 2021, 8, 2003185].

My main concern is that the results of this study strongly overlap those published in the literature [27-31, Appl. Catal. B 241, 89 (2019), Electrochim. Acta 326, 134976 (2019)]. The other serious flaws are found. 

(1) Figures 13-20 are very similar and show a negligible effect of the external strain on total DOS and minor changes in the MoSe2 part of the interface. Comparisons with the other MoSe2/Ti3C2Tx and MoS2/Ti3C2Tx should be provided. 

(2) Lines 629-631: "When there is no strain, the single-layer MoSe2 maintains its original semiconductor properties" Fig 5 or 6 should be cited here. Band structure with orbital contributions should be shown to indicate the band gap. 

(3) Figures 21 and S4 shows partial DOS of MoS2 in MoS2/Ti3C2Tx and MoSe2 and MoS2/Ti3C2Tx. The effect on total DOS is not discussed and shown, in particular, for MoS2/Ti3C2S2 and MoS2/Ti3C2Cl2 under different electric fields. It looks like the Fermi energy shift from one side of the band gap to the other. Total DOS should demonstrate the shift of VBE and CBE in the heterostructures.

(4) In Fig. 22d the point for zero electric field does not match the other points. 

(5) Line 743: "but also are rather sensitive to external strain and electric field." is not supported by the results, see, e.g., Fig. 13.

The results do not seem novel and substantial enough to warrant a publication in Nanomaterials. My recommendation is "reject". 

Reviewer 2 Report

This manuscript of Zheng et al. investigates systematically the effects of external field and novel functional groups X (S, Se, Cl, Br) on the structural and electronic properties of TMDs/Ti3C2X2 heterostructures using the density functional theory (DFT). The results reveal that the lattice parameters and interlayer distance of TMDs/Ti3C2 increase with the addition of functional groups. Both tensile and compressive strain obviously increase the interlayer distance of MoS2/Ti3C2X2(X=S, Se, Cl, Br) and MoSe2/Ti3C2X2(X= Se, Br). Moreover, this study provides insight into the combined effects of external field and novel functional groups on the related properties of TMDs/Ti3C2X2. Thus, the content as well as the topic of this paper is interesting for the current studies in the field and valuable from the theoretical research aspect. Unfortunately, the conclusions do not clearly cover and explain the obtained results which makes it difficult for reader to understand. Therefore, a major revision of the points given below is required:

  1. Authors should highlight the differences between this work and the previously reported works in the related fields.
  2. The manuscript text is too long and contains too many figures but lacks focus. This makes it difficult for the readers to grasp the main information. The authors should take care to highlight the main objective and the importance of this work clearly.
  3. Line 217-220: “It is noted that the interlayer distance d of the MoS2/Ti3C2 heterostructure is very small (in the range from 1.68 Å to 2.47 Å), which indirectly indicates a strong interaction between the layers.” Reference should be provided.
  4. Line 351-355: “From the partial density of states in Fig. 3 (b) and Fig. 4 (b), it can be seen that the electrons density of the Mo 4d orbital in the MoSe2/Ti3C2 heterostructure at the Fermi energy level is much higher than that in the MoS2/Ti3C2 heterostructure, indirectly indicating that a large number of electrons are transferred from Ti atoms to Mo 4d orbital in the MoSe2/Ti3C2 heterostructure.” It is not clear how the authors concluded that a large number of electrons transfer from Ti atoms to Mo 4d orbitals in the MoSe2/Ti3C2 heterostructure. Authors can provide more information to make it clear for readers.
  5. Line 442-445: “The interlayer distance of MoSe2/Ti3C2 heterostructure shows fluctuating change with increasing biaxial tensile or compressive. The fact may be that the MoSe2/Ti3C2 heterostructure has van der Waals interactions rather than strong electron coupling interactions.” The authors have given no reference to support this statement. It is difficult to understand how the authors confirmed (Figure 7b) that the MoSe2/Ti3C2 heterostructure has van der Waals interactions instead of strong electron coupling interactions.
  6. Fig 8a and Fig 8b show the nature of the fluctuation curves for MoS2/Ti3C2X2 (X=Se) and MoSe2/Ti3C2X2 respectively, which are not properly interpreted.
  7. There are many conclusions without any well-established reference. References should be included at appropriate places to strengthen the assumption or explanation more understandable to the reader.
  8. How can these first-principles study of TMDs/Ti3C2 heterostructures help for electrochemical energy storage applications such as batteries, supercapacitors etc.?  Please explain.
  9. More suitable references should be included in the appropriate places where any assumption is left open. Some related works based on MXene can be suggested to be included in the related places to support the importance of this work, such as: ACS Appl. Nano Mater. 2022, 5, 2470−2475; ACS Appl. Nano Mater. 2022, 5, 2358−2366; Batteries 2023, 9(2), 126 etc.

Reviewer 3 Report

In my opinion, the subject of the manuscript is actual. The authors address the issue of structural and electronic properties of TMDs/Ti3C2X2 heterostructures using the density functional theory. The authors analyzed the effect of various functional groups on the geometry and electronic characteristics of heterostructures, and studied the effect of an electric field and mechanical stretching. Generally, the work is fine. The chosen methods of computer simulation are adequate for the considered problem. The Authors describe the modeling technique in detail. In this way, the reader has the opportunity to double-check all the results, if desired. Also, this is very useful for further studies. The study is seemed interesting to me. Moreover, the manuscript contains interesting results for the theoretical and experimental science community. However, I have some queries that the authors should clarify and correct before the publication.

 1. First of all, I recommend the Authors to provide the coordinates of the optimized structures in Supplementary Materials.

2. When performing calculations, one should always be careful to select an appropriate k-points sampling and cutoff energy. Did the Authors check the convergence? For example, the Authors choose the value for cutoff energy ~33 Ry. At first glance, this may not be sufficient.

3. For the proper description of the heterostructures studied, it is necessary to take into account the van der Waals forces. The Authors perform their calculations considering Grimme’s D2 approach. However, D3 approach improves the accuracy of D2 through the use of environmental dependent dispersion coefficients and the inclusion of a three-body component to the dispersion correction energy term. So, the use of DFT-D3 seems more reasonable in the study presented. In addition, the computer cost of the D3 correction inclusion is negligible compared with the overall cost of a DFT calculation. So, why was the D2 approach chosen instead of D3?

4. The Authors should explain why they limited themselves to only biaxial mechanical deformations and did not consider uniaxial deformations.

5. What was the reason for the choice of the supercells? Did the Authors remake the calculations with larger supercells in case of adding different functional groups?

Round 2

Reviewer 1 Report

The authors thoroughly modified the manuscript according to my criticisms and suggestions. The manuscript was carefully revised. Also a professional English revision was done. The conclusions are now supported by the results, the novelty of the research is defined and added to the text. It can be published in the present form. 

Reviewer 2 Report

I have requested the revision and modification of Fig. 8a and Fig 8b. The authors have deleted these figures in the revised version with the argument that the results at these curves are fluctuating and therefore hard to describe them clearly. Similarly two tables and one more graph has been deleted instead of being revised. Due to these removals without any replacement, the manuscript content has been drastically reduced making it less worthy for publication. In this new version, there is a substantial deficit in scientific argumentations of the own results as well as with those in literature.

Round 3

Reviewer 2 Report

Thank you for the revisions.